# "Alone, there is nobody": A qualitative study of the lived experience of loneliness in older men living with HIV

**Amanda Austin-Keiller**[1]◉, **Melissa Park**[2,3]◉*, **Seiyan Yang**[3], **Nancy E. Mayo**[2,4], **Lesley K. Fellows**◉[5], **Marie-Josée Brouillette**◉[6,7]◉

1 Department of Psychiatry, Faculty of Medicine & Health Sciences, McGill University, Montreal, Quebec, Canada, 2 School of Physical and Occupational Therapy, Faculty of Medicine & Health Sciences, McGill University, Montreal, Quebec, Canada, 3 Lady Davis Institute, Jewish General Hospital, Integrated University Health and Social Services Centres-West Central Montreal, Montreal, QC, Canada, 4 Center for Outcomes Research and Evaluation (CORE), Research Institute of the McGill University Health Center, Montreal, Quebec, Canada, 5 Department of Neurology & Neurosurgery, Montreal Neurological Institute, McGill University, Montreal, Quebec, Canada, 6 Department of Psychiatry, Faculty of Medicine & Health Sciences, McGill University, Montreal, Quebec, Canada, 7 Research Institute of the McGill University Health Center, Montreal, Quebec, Canada

◉ These authors contributed equally to this work.
* melissa.park@mcgill.ca

**Data Availability Statement:** All relevant data are within the paper. The data need to remain de-identified and the supervising author and researcher who designed the study notes that "The

## Abstract

Loneliness has been shown to be a predictor of poor health and early mortality in the general population. Older men living with human immunodeficiency virus (HIV) are at heightened risk of experiencing loneliness. Here, we aim to describe the lived experience of loneliness in older men living with HIV and identify targets for intervention. We used grounded theory with a theoretical framework of narrative phenomenology to focus data collection and analysis on significant experiences related to loneliness. Based on individual narrative interviews with 10 older men living with HIV, experiences of loneliness related to "multiple losses," "being invisible" and "hiding out" as emergent themes. Participants also described living with loneliness by "finding meaning," "creating social experiences," "pursuing interests and things to 'live for'" and attending events in which "everyone is welcome." The discussion situates experiences of loneliness within the accumulation of losses and stigmas over time and how the participants strategies for living with loneliness could inform interventions to reduce loneliness in older men living with HIV at individual and societal levels.

## Introduction

Recent years have seen an intensified interest in the detrimental health effects of the loneliness. The evidence supporting the negative effects of loneliness on both mental and physical health is so compelling that the United Kingdom launched a cross-government strategy, "*A connected society: A strategy for tackling loneliness–laying the foundations for change*" in 2018 [1] while the 19th Surgeon General of the United States wrote a book about loneliness as a public health

number of participants is small and I would be concerned about potential breech of confidentiality."

**Funding:** This project was supported by grants from the Canadian Institutes of Health Research (TCO—125272 and HAL-157987) and the CIHR Canadian HIV Trial Network (CTN 273). None of the funding agencies played any role in the data collection, analysis, interpretation or design of this study.

**Competing interests:** The authors have declared that no competing interests exist.

concern [2]. In Canada, the Positive Brain Health Now study found that 64% of 836 participants aging with human immunodeficiency virus (HIV) reported feeling lonely sometimes or quite often [3]. This pattern is not surprising as, since the advent of effective antiretroviral therapy, more adults with HIV now live into old age which is itself associated with a high prevalence of loneliness [4]. In fact, earlier research identified loneliness as a frequent source of emotional distress that negatively affects quality of life among people living with HIV [5, 6], with 57% of HIV-positive adults over the age of 50 experiencing loneliness [7] compared to 35% of adults over the age of 45 in the United States [8]. Thus, understanding and addressing loneliness in those aging with HIV could improve quality of life, an important treatment target reflective of person-centred chronic care [9], overall health in general, and extend lifespan for persons living with HIV [7].

While there are many medical and community services available to people living with HIV, the experience of loneliness has not been directly addressed. For example, a 2017 survey conducted among a large representative sample of HIV+ individuals in the UK found that the greatest unmet need was related to social welfare, with a reported lack of available services focused on loneliness and-or isolation [10]. In 2018, Golden Compass, the HIV and aging program in San Francisco also identified social isolation as an key aspect of care to be addressed [11]. Yet, several ongoing gaps in knowledge continue to hinder the development of effective interventions.

Research on HIV and aging, as well as on HIV and loneliness, has been dominated by quantitative studies. Largely cross-sectional, the research often reports on the associations between loneliness and depression, stigma, physical symptoms, social support and use of alcohol/tobacco [12], leaving a knowledge gap on how persons living with HIV experience loneliness from their own perspectives. In addition, commonly cited definitions of loneliness used within current research represent third-person perspectives developed over two decades ago. For example, studies frequently cite the sociologist Perlman and Peplau's 1981 [13, p. 31] definition in which "loneliness is the unpleasant experience that occurs when a person's network of social relations is deficient in some important way, either quantitatively or qualitatively," while loneliness questionnaire are often based on/informed by de Jong-Gierveld's 1998 [14, p. 73–74] definition:

> Loneliness is a situation experienced by the individual as one where there is an unpleasant or inadmissible lack of (quality of) certain relationships. This includes situations in which the number of existing relationships is smaller than is considered desirable or admissible, as well as situations where the intimacy one wishes for has not been realized. Thus, loneliness is seen to involve the manner in which the person perceives, experiences, and evaluates his or her isolation and lack of communication with other people.

These earlier conceptualizations of loneliness do not take into account variance of experience across cultures [15], further limiting the generalizability of any related study findings.

Qualitative research can fill these knowledge gaps. The high prevalence of loneliness and its association with early mortality in the general population [5] and persons with HIV [6], overall negative impact on quality of life, the lack of services specifically designed to address loneliness for people living with HIV provide an impetus to better understand the lived experience of loneliness among older people living with HIV from their first-person perspectives. As the reported prevalence of loneliness differs across different age groups and genders [16, 17], this study focuses on older men one of the largest groups of HIV+ individuals in the Canadian context. The aims of this study are to describe their experiences to make recommendations for the development of services.

## Methods

### Qualitative approach

As described by Charmaz [18], the first-author took an inductive approach to grounded theory when coding for themes in line with a narrative-phenomenological theoretical framework [19] to focus data collection and analyses on significant experiences. Grounded in hermeneutic philosophy and narrative theory [19–21], narrative phenomenology foregrounds understanding the meaning of particular experiences of particular persons from their first-person perspective. As *both* narrative *and* phenomenological, this theoretical framework can be used to guide data collection around eliciting stories [22] to understand what is at stake or really matters for particular persons [23].

### Sampling

Participants were recruited from the Positive Brain Health Now (+BHN) cohort of 856 participants from five HIV clinics across Canada between 2013–2018, a comprehensively characterized research project with a focus on mental and cognitive health [24]. We used theoretical sampling such that the study participants were selected among the +BHN participants for theoretical relevance to the research question rather than for representativeness of the population [25]. For example, in order to compare and contrast the narratives of those with and without loneliness, we selected participants who reported feeling lonely either "quite often" or "almost never" on a question from the Older Americans Resources and Services [26] that had been completed prior by all +BHN participants.

For the study on the experiences of loneliness in older persons with HIV, participants were considered 'older' if they were 50 years of age or above. Participants were recruited and interviewed until theoretical saturation was reached, meaning the point at which no new information arose from additional interviewing [25]. Saturation occurred at a total of 10 participants who self-identified as "white." Six reported being lonely and four did not report being lonely in the survey. During the interviews, the participants spontaneously provided their ages, their work and living situations, and sexual orientation; a few also mentioned how long they had been living with HIV. As seen in Table 1, the ages of the participants ranged from 54 to 66; with 6 aged between 54 and 60 years and 4 aged over 60 years. Seven of the participants were not working and eight lived alone. They all self-identified as gay or bisexual with some

**Table 1. Participant demographic information.**

| | Participants | | | | | | | | | |
|---|---|---|---|---|---|---|---|---|---|---|
| | 1 | 2 | 3 | 4 | 5 | 6 | 7 | 8 | 9 | 10 |
| Lonely[a] | | x | | | x | x | | x | x | x |
| Age > 60[b] | x | | | x | | x | | | | x |
| Working[c] | | x | | | | | | x | x | |
| Living with someone[d] | x | | x | | | | | | | |

[a] The participants were asked "Do you find yourself feeling lonely: squares with an "X" represent the participants that responded, "quite often" (n = 6) and the blank squares represent the participants that responded "almost never" (n = 4).

[b] The squares with an "X" represent those that are over 60 years old (n = 4) and the blank squares represent those that were between 54 and 60 (n = 6).

[c] The squares with an "X" represent participants that were working

[d] The squares with an "X" represent the participants that were living with someone

identifying as long-term survivors while others had been living with HIV for 5 years or less. All ten participants signed formal consents approved by an institutional board of ethics.

## Data collection

Data collection proceeded in two steps: individual interviews and dyadic interviews.

## Stage 1: Individual interviews to understand experiences related to loneliness

Participants each completed an in-person narrative interview of two hours in the language of their choice (i.e., English or French). The interview guide consisted of four broad questions designed to elicit significant experiences or events related to loneliness in the participants' everyday lives from their perspectives, such as: "Can you tell me about your day yesterday?", "Can you tell me about the last time you really enjoyed yourself?", "Can you tell me about a moment when something got in the way of you enjoying yourself?", "Can you tell me about a moment when you felt lonely?" Prompts, such as, "Can you give an example," encouraged elaboration on the reported experiences (e.g., see Mattingly & Lawlor, 2000). Interviews were audio recorded, de-identified as they were transcribed verbatim, and transferred into MaxQDA for data analyses. The following conventions were used in transcription:

| -, | Participant paused and then continued with the previous topic |
|---|---|
| -. | Participant paused and started on a new topic |
| . . . | Participant paused for a long time. |
| [. . .] | Part of the quote was removed |

**Stage 2: Dyadic interviews to conduct member checking of emergent themes.** The individual interviews were complemented by member checking to add credibility to the results [27]. All study participants were invited to attend a group session to provide feedback on the emergent themes as analyzed from the Stage 1 interviews (e.g., validate or modify). Four participants expressed an interest in providing feedback, joining either a dyadic interview in French or a dyadic interview in English. Six participants were either not comfortable with the group format or were not able to participate in Stage 2. Each session lasted over three hours, during which the participants responded to the themes with experiences that confirmed or refined them without the interviewer interjecting. At the end of the session, the researcher integrated a participatory component into the session by asking participants about the type of services that they would find helpful to redress loneliness.

## Data analyses

**Stage 1: Individual interviews.** The transcripts were analyzed in their original language (French or English) so as not to lose any meaning in translation. Thematic analyses of the data followed several steps: initial coding, focused coding and theoretical coding [18, 25]. For initial coding, each segment of data was named with a code derived from the data (e.g, *hard to make friends*), as opposed as being named from preconceived notions (e.g., *socializing*). A second round of coding, called focused coding, used the most frequent codes established in the initial coding (e.g., *hard to make friends + rejected because of HIV + rejected because of old age*) and clustered them together into categories based on commonalities (e.g., *challenges making friends related to others' perceptions of HIV and aging*). A working hypothesis was formed from these

broader categories (e.g, *negative attitudes related to HIV and aging + fear of facing more negative attitudes*), which is called *theoretical coding* (e.g, *loneliness related to negative attitudes about HIV and aging + fear of those attitudes*). The emergent themes were further modified following the analyses of each interview (i.e., the example presented above was developed after many rounds of interviews) and were presented to the participants during the Stage 2 dyadic interviews for member checking.

**Stage 2: Dyadic interviews.** Based on the feedback from the participants, the final theoretical codes and emergent themes found in Stage 1 were refined in the following manner. First, we looked for congruency between the Stage 1 theoretical codes and what the participants discussed during Stage 2. AA-K, MP and M-JB then drew more heavily on narrative phenomenology to identify the significant experiences and events related to the theoretical codes, for example the use of present tense or metaphors while talking about past events, heightened emotionality [22]. These moments can also include insights into the larger cultural, historical and social discourses within particular contexts that persons use to make sense of their experiences [19], for example, heightening attention to the relationship between the accumulation of losses, negative attitudes of others, and experiences of loneliness. The theoretical codes were then discussed by the authors who each drew from different epistemologies related to their different backgrounds in psychodynamic psychotherapy, anthropology, psychology and phenomenology to provide further rigour to the analysis. To illustrate the final codes, exemplary quotes were drawn from the individual and dyadic interviews to represent the final codes. All quotes were translated into English and nominal information was removed to preserve confidentiality.

The analyses of the dyadic interviews as a form of member checking refined the final themes, for example, to foreground the participants' experiences by renaming the codes from the participants' 1st person perspectives rather than 3rd person researchers' perspectives. For instance, a code initially labelled, "loss of a significant other" was changed to "multiple losses" to better reflect the participants' experiences. The member checking process led to the refinement of researcher codes, for example, "hiding oneself" became "hiding out" and "being invisible" to reflect the participants' experiences of hiding their sexuality and HIV status as well as feeling as if they were not seen or acknowledged by others. Similarly, more in-depth narrative analyses of the experiences previously placed under the code of "distracting oneself" was changed to "pursuing interests and things to 'live for'" as more aligned with the actual intent or purpose of engagement in different activities using participants' own terms.

## Results

**Table 2** provides a visualization of the themes as they emerged in each of the interviews and the point at which saturation was reached. The quotes presented in the results are all from Stage 1, the individual interviews, unless otherwise specified as being from Stage 2.

### Experiences of loneliness

Three major themes emerged as experiences related to loneliness: "multiple losses," "being invisible," and "hiding out." As can be seen in **Table 2**, these three themes were discussed by all 10 participants. In conjunction with these themes, each participant described moments when they faced the negative attitudes of or rejection by others because of sexual orientation, HIV status, or aging.

**Multiple losses.** During the interviews, all the participants spoke about loss. The participants talked about their partners and close friends who had passed away from HIV, how these

**Table 2. Emergent themes.**

| | EXPERIENCES OF LONELINESS | | | LIVING WITH LONELINESS | | |
|---|---|---|---|---|---|---|
| | **Multiple Losses** | **Being Invisible** | **Hiding Out** | **Finding Meaning** | **Creating Social Experiences** | **Pursuing interests** |
| 1 | X | X | X | | X | X |
| 2 | X | X | X | | X | X |
| 3 | X | X | X | | X | X |
| 4 | X | X | X | | X | X |
| 5 | X | X | X | | X | X |
| 6 | X | X | X | X | | X |
| 7 | X | X | X | X | | X |
| 8 | X | X | X | | X | X |
| 9 | X | X | X | X | X | X |
| 10 | X | X | X | X | | X |

Data saturation grid: the rows represent the participants [from 1 to 10] and the columns represent the themes that emerged: a square with an "X" indicates that the theme was discussed by the participant.

deaths made them feel very alone, and the various ways being alone manifested in their daily lives. For example, Paul explained how much he misses regular contact with another person:

> Love. It's the contact. I am not talking about the sexual side. I am talking about talking to a person, touching a person, something like that [. . .]. Alone there is nobody.

He continued by explaining how the "*little things*," such as sharing meals, add up as concrete daily reminders of what is missed:

> I miss the little things. It's true that when you live alone, there are little things that you miss. Just a touch or-, I don't know, just another person there. Eating alone is really boring-, you know, I always eat standing up in the kitchen. I have a small dining table, but I prepare it in the kitchen and then just eat at the counter, standing.

All the participants reflected that the sheer magnitude of their losses had been missed. They had lost multiple people at a young age, many more than what most people experience. For example, in his individual interview Eric recounted:

> In the years '80-'85, I met people. I met couples. But when the disease came, 90% of the people disappeared. I lost more than 90% of my friends.

Charles not only shared the multiple losses he experienced from HIV-related deaths, but also hints at the qualitative impact on a generation of persons:

> You get to know someone really well and they would die because this was the era that people would die, and many, many people-, [. . .] so it was difficult sometimes because you get close to somebody and then they would pass away.

These are the losses, not of mere acquaintances, but of "many people" with whom they were "close." For Charles, this is the loss, to cite Paul's experience mentioned above, of "*touch, of just another person there*" on a scale that can only be represented as an "*era*."

In addition to losses resulting from the death of others due to HIV, participants also described the loss of friends and-or family when they disclosed their HIV status. For instance, when Ben invited people he had met at a community group for people living with HIV to dinner, his friend decided not to come:

> I was always thinking that she was open open [. . .]. One day she called me and said, "Oh you and your guys and girls from that group [. . .] I don't want to meet you anymore and eat at the same table. I'm going to catch something". I said, "What? Are you crazy, lady?" But maybe she thought about that for a long time, and she never had the guts to say it and one day, "boom," she decided to say it.

This story emerged during the dyadic interviews for member checking, shifting a previous theme of "loss of a significant other" to "multiple losses," to better capture the accumulation of losses whether due to death, rejection or the public image of persons with HIV as being only "skeletal beings on their deathbed."

Losses were also related to when participants, themselves, first learned about their HIV status. Frank, for example, recounts an accumulation of losses that begins when he learns about his status:

> I found out I was seropositive. I lost my job. I didn't have a salary. I thought my life was over, my career was over. So that is how it developed. My partner was sick for 10 years, he had heart problems, he was at risk for sudden death. I was in survival mode.

Frank's loss of his job, his salary and a vision of the future accentuates the loss of any certainty that had pervaded his life with his partner who "*was at risk for sudden death.*" He begins to drink, which leaves him even more isolated:

> I was isolated but I was also isolated because of my own difficulties. Because I was drinking. It brought on more isolation. Not only the HIV but my own difficulties and my behaviour resulted in that. And it results in isolation because when you have weird behaviours, you have problems. You have traumas. You react in a certain way and the people that are doing fine-, they go in the other direction.

Frank's experience of being in "*survival mode*" is a state in which one can only "*react*" alludes to both ones own loss of a sense of living as well as others who "*are doing fine-, they go in the other direction.*"

**Being invisible.**   The participants described the barriers they faced when trying to expand their social circle to find the intimacy they were lacking. In the analysis of these barriers, the emerging theme was how being older and having HIV made them feel invisible. The participants described continuing to experience loneliness despite trying to meet people and develop new social relationships in person or online. For instance, participants explained that the gay bars, which they often frequented in the past to socialize, no longer worked as effectively to meet people. As an illustration, Sebastian described his experience as follows:

> As you get older I notice that,—it's like nobody is looking at me anymore, "Okay, I am invisible." That is what it feels like-, but, you know, age is a factor for sure. Straight or gay. [. . .] Age is a big thing for any person to deal with and you just have to live with it-, and find your place.

Age was also factored in dating apps. For instance, Adam explained:

It's one thing to be online when you're cute and 26. And it's a totally different thing to be online with lots of photos when you are 58 and everybody is looking for cute guys who are 26.

Although "being invisible" was a theme that emerged in relation to age, this theme also included experiences related to the disclosure of one's HIV status

For example, when using an app, some participants opted to keep their HIV status invisible or private to increase the likelihood of arranging a meeting with someone. However, this only postponed confronting the barrier related to HIV disclosure. Adam explained that if he kept his HIV status private on the app, he could more easily arrange a meeting with someone. Yet, he then must grapple with when and how to disclose.

So the HIV disclosure. I've tried over the years. I've tried all sorts of methods [. . .]. It makes for a really lousy first date-, and then the guy he-, you can see him kind of going off in his head-, and he's talking, and he's gone and he's far away-, "Okay . . . Well . . . So I guess we'll see if he ever phones again," kind of thing.

Overall, the participants described experiences of loneliness in which the visibility of their older age and-or HIV status left them "being invisible" to others when searching for long-term and intimate relationships.

**Hiding out.** The participants described actions to isolate themselves. Emotionally as well as physically, "hiding out" becomes a shield against the pain caused by experiencing or witnessing barriers related to HIV. This active "hiding out," however, follows an initial period in which loneliness is first caused when HIV shifts one's relation to society. As Paul describes:

Living with HIV and loneliness, I think it is common. Because whether you like it or not, your life is flipped upside down [. . .]. It's there that the distances form-, the barriers and the loneliness also-, I think it comes with that.

Paul exemplifies the distance and barriers associated with loneliness and HIV as akin to what is experienced with a diagnosis of cancer, in which people withdraw from someone because they do not know what to say and how to react:

Like when you tell someone you have cancer. At one point the person withdraws. And I think that is normal too. It is part of life.

However, he goes on to describe how a HIV diagnosis is worse than cancer because it is "*badly digested by society*:"

And with HIV it is even worse because it was badly digested by society. So automatically there is a form of protecting yourself. Withdrawing. The least you appear, the least they can see you. That's what I saw.

This is an active process of "hiding out," in which "*the least you appear, the least they can see you.*"

Notably "*withdrawing*" as a form of "*protecting*" oneself often started early in life, before any HIV diagnosis. For example, Adam describes how "*hiding out*" started in school:

My overriding goal at school was to do everything I could, not to be bullied. So that created very much a kind of hiding out-, "Don't be noticed." And that-, one of the recurring things that I haven't been able to shake, like from the growing up years of umm-, you know, "Don't-," certainly, "Don't be noticed for your sexuality. Try not to do anything too feminine," you know, you want to be under the radar at all times.

In contrast to being invisible, the practice of being "*under the radar at all times*" was purposeful, an active "*hiding out*" from society related to one's sexual orientation or how one's gender might be perceived by others; that is, "*to not to do anything too feminine.*"

There was a prevailing homophobia around, so I was very aware of that from kids at school and name calling and things like that. And whenever I observed bullying, I thought, "Oh it won't take much for me to be next." So that-, um you know. . . I really integrated that.

Adam then underscores how the impact of the bullying he witnessed as a child continued to impact on every facet of his life:

You know, I don't really like that expression, "internalized homophobia," but I recognize what it means-, but certainly I have it and had it-, . . . and it affects you just about everywhere and it affects your decisions-, you know, your actions and your decisions.

Overall, the repeated negative experiences related to HIV status or sexual orientation reinforced the participants' desire to avoid such situations by "hiding out," which contributed to further experiences of isolation and loneliness.

## Living with loneliness

The participants used many descriptors for living with loneliness. Although one could live with loneliness, at other times the experience was that one had to "survive," "deal," "cope," and/or "get help" with loneliness. The participants' stories also brought up the range of actions in which they diminished their experiences of loneliness or its effects by finding meaning through spirituality; creating social experiences through animal companions or other, more brief, encounters; and pursuing interests and things to 'live for' in shared events.

## Finding meaning

The participants discussed their beliefs or spirituality or, in their words, "*believing in something greater.*" For instance, as Sebastian described:

Like everybody believes in something. Whether you call it God or any other religion-, something that we are all supposed to have-, this faith in something. Otherwise, there is no purpose.

None of the participants mentioned, however, following any specific organized religion. In fact, for the participants, organized religion harbored hypocrisy. For example, as Sebastian explained:

The Church has been a huge problem for me. Especially coming out as gay. It is like, "Why are you spreading all this hypocrisy?"

Yet, participants did discuss that having a belief in a greater being and/or destiny, ultimately allowed them to come to terms with their HIV infection and not blame themselves. For example, Frank described how his belief in something greater helped him foster self-love and acceptance:

> If you believe in something greater than you-, it could be the chair, you stop fighting yourself. You have confidence-, it's-, to develop confidence in something greater than you-, that can change you, without really changing. But if you stop fighting yourself and you come to accept yourself how you are, you love yourself.

Frank distinguishes being alone from loneliness, explaining that one cannot experience loneliness when one is "*loved by something greater*," no matter what one calls it:

> There is being alone and then there is feeling lonely. But when you have a deep conviction that you are loved by something greater than you-, by life-. Anything. There is no loneliness.

Beliefs in something greater also helped some participants deal with loss; for instance, Sebastian, spoke about how his spirituality helped him deal with others "leaving" and when they "pass on:"

> Seeing all the other people suffer and then leaving. It has helped me in a way-, . . . as-, . . .as- . . . what I have always thought about the afterlife-, what is going to happen. And for me, it is seeing these people pass on. I almost come to believe that you can actually see the person's spirit leaving their body. [. . .] Everything that we accumulate here. . .survives somehow-, so I call it energy. I can't really explain it and never will, but it helps in dealing with all that and to say, "Well there has to be a purpose."

Overall, "finding meaning" related to something greater than oneself, even "*anything*," or a purpose supplanted the experience of loneliness brought about by multiple other losses.

**Creating social experiences.** A number of participants had cats or dogs as pets, describing how they provided companionship, unconditional love, and social experiences. For example, Jack recounted how his cat helped him through a difficult period:

> I have a cat. I can't get rid of it because it's the only thing that helped me through that period. He's a lovely pet. [. . .] He chose me and I chose him [. . .] it's like a link, too, with my boyfriend.

Jack links the quality of his relationship with his cat—the active choice on both of their parts—to the quality of relationship he had with his boyfriend as much as being linked to memories of their being together. Domenic, another participant, mentioned how his dogs gave him a reason to get out of the house and helped him create connections to others, even if momentary.

> First, people find me good-, having dogs, having the discipline of four walks a day and by doing these walks, especially summer or spring, I walk in the Village-, I walk in the street-, I say "hi" to people and people say "hi" to me. There can be people that don't know me, but we can have an exchange. We can talk maybe 10 minutes, thanks to the dogs. I won't see

these people again, but I had an exchange with them. So, it comes and helps with the loneliness.

Even more than these brief exchanges, Domenic's description also suggests how the activity of walking the dogs also provided a type of character reference for others as to his goodness or discipline.

In contrast, other participants tried to create social interactions by using dating applications. Sometimes, these interactions were equally limited to brief exchanges. For instance, as Ben, explained:

The cellphone . . . it's rapid sex. It's a meat market, "Oh he has a big dick. Oh yes, I'm home. Are you free?" So, you go over there, and you don't want to connect. You just want to have sex.

Unlike Dominic's every day, four times a day, experience of brief exchanges with others while walking his dogs, Ben's encounters left a void: "*So, you just have sex and they leave and after you end up with yourself.*" Frank also commented on the limits of brief sexual encounters to counteract loneliness, stating, "*but to break the loneliness, I had tried that [sex]. And I was doing well with it, but it was superficial.*" The loneliness was only broken for the time that Frank was with the person and remained "*superficial.*"

In the context of creating social experiences, drug use was mentioned by some participants in terms of enhance the quality of the experiences. As Charles recounts:

I do dabble with something on Sunday afternoon to make you a little more fun, but not in any kind of abusive way-, but-, you know, if that is the only time when you feel like you can be social and happy.

Drugs are about making one "*a little more fun.*" Yet, Charles' reference to doing drugs as "*the **only time** when you feel like you can be social and happy* [emphasis added]*"* hints more about how his everyday life is dominated by isolation and unhappiness.

**Pursuing interests and things to "live for".** Most of the participants reported that when they felt lonely, they tried to do things. For example, Charles describes how he draws, paints or goes for walks:

I get bored. I draw. I paint a little bit too. My hobbies. But. . . sitting around thinking, "Okay . . . I am really depressed." It is easy to do but it is not me. It is just always something I can do, but whether it is just in my 3 ½ [small apartment] or going outside for half an hour walking around, just to change my mind a little bit.

For him, activities were a way of not letting his mood become negative, "*But not sitting down and letting it [my mood] go down. That is the worst thing that you can do. So, no, no-, I stay clear of that.* For others, doing things is a form of prevention. As Gregory recounts:

Well, for me to prevent loneliness, I just keep myself really busy. Like, I am very active-, like I try to pursue some interests like, you know, vegetarian cooking and, you know, self-absorbed interests. But who cares?

Overall, the participants explained that doing something and keeping busy provided an interim solution to "stay clear" of loneliness. However, busyness by oneself did not always

remedy a prevailing experience of loneliness. The most memorable times were those when one's personal interests were shared within larger public events. For example, Charles described his experience at a music festival, which he really enjoyed because he could have fun with everyone that was around:

> The last time that I really enjoyed myself was,- I would say it was a couple of weeks ago . . . we went to a techno festival and it was the first one, you know-, I was really looking forward to it because it was the first one and we kind of live for it. [. . .] An unexpected joyous time, where we were really happy-, we were-, I was crazy for the music and it was so much fun. [. . .] We had a really good uplifting experience.

Charles described how the joyous time was "*unexpected*" and a "*really good uplifting experience*" when what is really matters was having and sharing "*so much fun*" rather than age or sexual orientation:

> We had a nice day with people that we had met a long time ago and we met with them and had so much fun and they were all in their 20s [. . ..] It's really fun because it is very reinforcing. Even though we are older, we can still have fun-, they liked us being there. They were not like, "Oh look at that old faggot." We prove that-, it doesn't matter. If you like the music, it doesn't matter what you are. It doesn't matter if you are gay or straight.

Charles' experience shows that the personal interests that he and his friends "*live for*" can also lead to connections with others in social events in which what is shared is much more important (and reinforcing) than age, sexual orientation or other markers of difference.

### "Everyone is welcome"

During Stage 2 dyadic interviews, the four participants described services they thought could help to reduce loneliness. Overall, they agreed that HIV community groups are instrumental for providing support to the newly diagnosed or to those who are having difficulties with medication or navigating medical or social services. Yet, they also reported a lack of groups for people who are generally doing well but are experiencing loneliness and seeking to develop long term platonic and/or romantic relationships. They also expressed wanting groups not limited to HIV-positive individuals. Specifically, during one dyadic interview Ben asked:

> Why a closed group? You know, you want to have an openness from everyone, so it should be opened. But like an HIV event, but everyone is welcome. Kids can come. Because one day they might face it if we don't cure that virus. You don't want to keep yourself just "Me, me, me, me".

By advertising that the events are for HIV-positive individuals with others welcome, they hoped that members of the general population would participate and that anyone uncomfortable with HIV would simply not join, creating an environment where "*everyone is welcome.*" The Pride parade is an example of such a welcome format and where it can be expected that people who attend are open and accepting of differences.

The participants suggested that groups could be centered around activities such as arts, sports, cooking and the like to allow for connections to be made based on similar interests as opposed to repeating support group formats, dating applications, or bars. Peer mentoring sessions were also of particular interest. During a dyad interview, Ben explained:

The new generation is coming for HIV and we should face them with the old generation so they can deal with them, all together: "Yes, you have that and that and that and now there are people that have lived 20 years already with it. They will tell you all about it." [. . .] It is something that no doctor will tell you.

The participants not only spoke about what kinds of services that could be developed now that would help them, but also how future generations could be nurtured. Two participants discussed the importance of education on HIV and sexuality for youth at school and with families:

Participant 1: But for education, at school. Are we teaching these things?

Participant 2: I have friend who worked, and he did the tours of the schools to talk about [HIV]. It's not easy to do.

Participant 1: The earliest we can, we need to give the information to the population.

Participant 2: And the person's social network is important too. Because if the family is problematic. . ..

Interviewer: What education specifically?

Participant 1: Sexual education.

Participant 2: There should be classes on socializing and sexuality.

Participants felt that topics such as prevention of sexually transmitted infections and safer sex practices should be included in school programs. The aim of the education would not be limited to providing necessary information for prevention of sexually transmitted infections but also to decrease the negative associations related to them in addition to more education on healthy relationships and socializing as they felt that children who most often use online means of communication have difficulties expressing themselves in person. Overall, they hoped that the education system could inform young people and dispel negative beliefs about sexuality to build a more inclusive society for future generations, not just for themselves.

## Discussion

The lived experience of older men living with HIV contributes original knowledge on their first per son experiences of loneliness and its relationship to an accumulation of loss(es), stigma and intersectionality. How these men describe and live with loneliness could situate and refine outmoded third-person definitions and related measures of loneliness, while informing service development. In addition, their experiences of inclusion and suggest effective strategies to address increasing public health level concerns about loneliness in general and loneliness related to intersectional stigma in specific.

### Loneliness and the accumulation of loss(es) and stigma(s)

All the participants in the current study described how they lost friends and partners to HIV, in addition to losing persons they had trusted right at a time when they really needed their support. The sheer number of these accumulated losses left as little as 10% of their social network remaining. Further, the "*little things*" they missed every day were constant reminders of those multiple, accumulated losses. Overall, the theme of "multiple losses" depicted the immense losses experienced in the context of the AIDS epidemic and continuing negative attitudes about HIV, which left participants alone and feeling lonely. As noted by Zeligman & Wood

[28], the loss of social networks also include the loss of what might have been previously desired or imagined, which is only further accentuated in societal contexts in which HIV deaths are negatively viewed or even go unacknowledged. The experience-near aspect of the current study also amplified the equally devasting losses of the "little things," such as "talking to a person, touching a person," and other shared moments during a day. The participants linked their loneliness to an *accumulation* of losses that built over time, from the loss of intimate others to the losses in an entire era, from the loss of work to the loss of the everyday "little things" of touch and presence and, ultimately the loss—or risk of losing—any sense of living at all to merely being in "*survival mode.*" This accumulation of loss(es) was inextricable from their experiences of stigma and its intersectionality.

The participants experience of multiple and intersecting stigmas—negative social attitudes attached to a characteristic of an individual that may be regarded as a mental, physical, or social deficiency [29]—threaded throughout their stories of loss, both directly and implicitly. For example, the accumulation of losses often started from a very young age with the loss of familial support due to their sexual orientation. Stigma related to homophobia compounded stigma related to their serostatus. In line with early research that individuals with "concealable stigmas" are vigilant and suspicious of anyone who might recognize what they are concealing and be more likely to avoid social situations [30], a more recent study of aging and HIV underscored how previous experiences of stigma or "anticipated stigma" can lead to avoidance of disclosing "HIV status as well as sexual orientation to others, including friends, family members and loved ones" and even "to avoid disclosure of their HIV status altogether [to potential partners], even if it meant entirely eschewing romantic relationships as a solution" [31, p. 140, 143].

Stigma around sexual orientation and HIV status are compounded by societal views on aging, itself. In general, older age is a source of rejection and name calling [32] and, as noted more recently by Goll et al. [33, p. 13], loneliness among older adults—even those living independently—is a prevalent and growing public health concern specifically when they avoid going out for fear of invalidating their preferred identities as "independent and youthful." The finding that even anticipation or fear of being rejected or exploited creates barriers to social participation only heightens attention to how the combined effects of stigma related to sexual orientation, HIV and older age contribute to "a shrinking kind of life" [34, p. 330]. Perhaps most alarming is how this reduction of life, itself, is reflected within the participants' experiences of "*being invisible*" in their own on-line dating community due to their age. As one participant had noted, "*nobody is looking at me anymore*" especially "*. . .when you are 58 and everybody is looking for cute guys who are 26.*"

As the sociologists Barraket and Henry-Waring [35] noted in 2008, online dating trends are both shared by and situated within a broader socio- cultural trends related to work, households, and mobility patterns. Over a decade later, Conner's [36] qualitative study of Grindr, designed for gay men to interact and connect with one another based on their proximity, found that the on-line platform reinforced existing cultural practices (e.g., body typing, ageism, racism) while also creating new ways for its users to stigmatize or marginalize others. This is best exemplified by how users of Grindr "create a dichotomy between HIV-positive and HIV-negative men, using the label 'clean' to describe HIV-negative men" [36, p. 409]. This knowledge was well understood by the participants in this study, who not only explicitly linked "being invisible" to their older age but also to when they disclosed their HIV status and then see the potential partner "*kind of going off [. . .] he's gone and he's far away.*" Although on-line platforms might provide new opportunities for interactions for "exploring sexuality on their own terms,"[37, p. 9], they also reinforced multiple stigmas.

The results of two studies from the +BHN cohort—from which the participants of the current study were drawn—found that both loneliness and stigma, considered separately, have widespread impact on brain health and quality of life [3, 38]. Further, Harris et al. [3, p. 366] reported that adults living with HIV are at increased risk for experiencing "loneliness with stigmatization and economic marginalization added to the health challenges arising from chronic infection." These results are also supported by findings of another quantitative study that loneliness mediated the association between stigma and depressive symptoms. In fact, stigma predicted higher loneliness which, in turn, predicted more depressive symptoms [39, 40]. The results from this qualitative study provided rich descriptions to illustrate the linkages between the experiences of loneliness, the accumulations of loss(es) and stigma at the intersection of sexual orientation, HIV status, and aging, in which the negative societal attitudes about HIV status and aging are also reflected within the dating community of older gay men.

## Living and loneliness

Participants described having agency in their life the ways in which their actions to live—in contrast to "*survival mode*"—with loneliness could inform interventions at individual and collective levels. Further, the participants provided examples of inclusive experiences and what helped to create inclusive experiences during member-checking that provide additional insights on the need for societal interventions and directly involving them in the planning stages.

**Being alone without feeling lonely.** Participants described how their spirituality or "*believing in something greater*" contributed to their capacity to be alone without feeling lonely, an indicator of emotional maturity. Research has established that the feeling of being in the presence of a greater being can decrease feelings of social deprivation [41]. Likewise, in the words of one participant, having a "*deep conviction that you are loved by something greater than you-, by life-. Anything. There is no loneliness.*" Spirituality also contributes to psychological well-being by mediating the negative effects of HIV stigma [42, 43]. In contrast, and similar to the results of a qualitative studies on LGBTQ experiences with religion and spirituality [44] and the transition to coming out gay [37], the participants marked the "*hypocrisy*" of organized religion and experiences of isolation based on their sexual orientation.

In the early literature, loneliness and sex in the HIV+ community was often discussed from a public health perspective in terms of risky sexual behaviours that could lead to transmission of HIV [45, 46]. Yet, our participants described experiences in which disclosure of HIV+ status led to the disappearance of others, while dating apps actually exasperated loneliness when, as the study by Conner [36] underscored, some users refer to people with HIV as 'dirty' and those who did not have HIV as 'clean.' Encounters were often fleeting, leaving participants feeling "alone" while the companionship of animal also helped participants live with loneliness. Pets did not just represent and link participants to former relationships, the pets were also mutually agentic relationships in which each had chosen the other. A systematic review of loneliness and animal companions concluded that companion animals were associated with less loneliness, with the reservation that the methodological quality of reviewed studies was questionable [47]. Our findings suggest that the importance of animal companions in loneliness warrants further investigation; since the participants also described how their pets also helped them create habitual social experiences by providing opportunities to go out and talk to people on a regular basis while also providing the a kind of character reference in terms of their "*discipline.*"

In this current study, participants also pursued individual interests to change their thoughts or prevent dwelling on loneliness. Of note, many of the hobbies the participants described

were individual pursuits, which potentially protected them from experiencing rejection or more stigma. Although, research suggests that chronically lonely individuals to tend to turn to individual activities to cope, paradoxically maintaining the loneliness [8], even the "*self-absorbed*" interests of participants offset depression. In fact, personal interests that a participant "*lives for*" connected him to a "*we*" in public (i.e. techno music), in which what was shared was more important than any differences defined by age or sexual orientation. Thus, "pursuing interests and things worth 'living for,'" even if through individual activities, also provided the grounds to foreground similarities rather than difference, which could ultimately help shift prevailing negative social attitudes and assumptions related to stigmatizing characteristics whether visible (e.g., age) or 'concealable' (e.g., sexual orientation, HIV status). At the same time, however, personal interests and activities as ways to redress loneliness are dependent upon inclusive spaces. As noted by Lukas et al. [37], even "queering" meaningful activities—maintaining activities in ways that align with gay identity, such as going to gay bars or joining gay sports teams or leagues—requires finding "safe spaces" and safer spaces are always fluid, dependent upon the attitudes of others within those spaces.

**What can we do now?.** In the literature, interventions for reducing loneliness often emphasize increasing access to social support [48]. Yet increasing access to social support does not address the underlying issues. For example, research shows that individuals who experienced multiple losses during the HIV/AIDS epidemic can develop post-traumatic stress-like symptoms, such as numbing, flashbacks, nightmares, emotional outbursts and substance abuse that, in turn, further impairs their ability to build new social connections [30, 49]. Although this research does not account for how embodied trauma and, more specifically, fear of its recurrence may be equally disruptive towards building new social connections, Frank—a participant in this current study—clearly noted a link between "*traumas*" and his increasing isolation. He then attributes his loneliness in terms of his "*strange behaviors*" and drinking. Yet, the findings from the current study also suggest how post-traumatic stress could also be linked to the accumulation of multiple and <u>ongoing</u> losses and stigma over time not limited to experiences that occurred during the HIV/AIDS epidemic.

In addition, as the study of older adults in general demonstrated, even fear of stigma is a barrier to social participation [33], and a meta-analysis of interventions to reduce loneliness reported that interventions that targeted maladaptive social cognition, such as the fear of experiencing stigma, had larger effect sizes than the ones that attempted to improve social support or those that increased opportunities for social support [48]. The participants in this current study were keenly aware of the repeated experiences of stigma have left them fearful to make social connections as they describe a balance between taking the risk to go out and seek enjoyable activities and "hiding out" to protect themselves from experiencing stigma. Directly identifying fear as a maladaptive reaction to stigma or using cognitive behavioural therapy to help overcome those fears might be a useful, even critical, addition to other interventions to address loneliness in this population. Yet, the participants in this study did not identify fear as an area for intervention.

Instead, based on their actual experiences and knowledge, the participants provided specific suggestions during member-checking for how to redress stigma, including using their expertise to educate children and youth in their schools or the need to develop appropriate (i.e., stigma-free) meeting venues. These suggestions are supported by a 2016 collaborative priority setting for research topics with persons living with HIV, health professionals and policy makers, in which the two highest research priority topics and examples were 1) public education to address stigma in the context of rehabilitation (e.g., reduce workplace stigma); and 2) impact of stigma on activity, participation, and rehabilitation service delivery (e.g., need for effective long term strategic interventions to reduce stigma) [50]. Likewise, Boggs et al. [51, p. 1539]

underscored the need to reframe the lived experience of stigma related to ageism, heterosexism, and/or cisgenderism as a resource: "Resilience from weathering a lifetime of discrimination was identified as a strength to handle aging challenges." In fact, even if older gay men often hid their sexual identify in health care settings, those living with HIV disclosed their sexual identities for both pragmatic and political reasons related to advocacy. Even those who had "positive and affirming experiences" with healthcare professionals discussed "anticipating the need to occasionally challenge stigma and discrimination" [52, p. 262].

Since stigma has been found to negatively impact access to healthcare, housing, social support, home assistance and legal services [51], societal interventions to redress stigma could have a more profound reach across different social determinants of health for all persons. For example, in Kottorp, Johansson, Aase & Rosenberg's [53] descriptive study of housing needs found only minor differences in preferences for senior housing between LGBTQ+ and heterosexual matched controls in terms of activity options, environmental features and staff competence. Although the LGBTQ+ older adults stressed the need for safety in the form of staff trained in LGBTQ+ issues, the values that emerged in the LGBTQ+ focus groups—such as viewing helping someone and being needed as opportunities rather than burdens, solidarity and support in everyday activities—could also create inclusive senior housing for others as well. Since residents would constitute an extended family of choice, a sense of safety would be extended to all. Similarly, the perspectives of LGTBQ+ older adults on aging in place recommend the need to establish welcoming communities [51].

Access to stigma-free environments would provide safe opportunities to not only establish new relationship, but also educate others merely by being present. For example, in the theme of "*Pursuing interests and things worth 'living for,'*" the participants described events such as a music festival in which they could socialize and enjoy themselves without feeling the threat of stigma. Under the theme, "Everyone is Welcome," the participants also verbalized that they these public events become critical to also show others that they are not 'skeletal beings on their deathbed' but aging well with HIV. In addition, the participants in our study emphasized that educating the youth was important to them to prevent further HIV infections and for stigma reduction. Anti-stigma campaigns could benefit from the lessons learned from another stigmatized condition, mental illness. For instance, The Mental Health Commission of Canada's *Open Minds* anti-stigma initiative used contact-based education, an approach that emphasizes stories of hope and recovery, as individuals who have experienced mental illness firsthand tell their stories to youth, healthcare providers, workplaces and media/journalists [54]. In addition, the Public Health Agency of Canada also funded *POZitivity*, a national HIV anti-stigma campaign which aimed to challenge stigma, through social media and the *Slay Stigma Canadian Drag Tour* [55]. Evaluation of these initiatives could inform further health promotion campaigns.

## Limitations and future directions

Qualitative research methods allow for the meaningful connections that participants make to be understood; we have reported the themes the way the participants narrated, as opposed to attempting to fit them within known, pre-existing themes. With this method, we can come to understand how the participants experience and live with loneliness. A limitation to the method is that only one person completed the initial coding. However, the themes were discussed and reviewed by all researchers, and member checking was performed with participants to refine the endorse the themes. Finally, themes were again refined and grounded in participants' own terms during the final hermeneutic analysis. Another limitation of this study was the sample size (n = 10) and no comparison group. Moreover, the study participants were all

older gay/bisexual white men living in Montreal, Canada. Stigma and the perspectives of other groups who are living with HIV (including straight men, women, members of other cultural groups or ages) are likely to be different dependent upon the complexities of social location, positionality and context [52]. In addition, the participants themselves mentioned that individuals living in more suburban or rural areas are likely to have more pronounced experiences of isolation, stigma and loneliness. Therefore, the needs of these populations may require further research.

Finally, the participants described using drugs to "*make you a little more fun*" and alcohol when they were in "*survival mode.*" This echoes the findings that cannabis use in HIV-positive individuals is in part motivated by its alleviation of stress and relaxation [56]. Yet none of the participants indicated that drugs or alcohol led to the alleviation of loneliness over time. Although many studies have investigated alcohol or drug-use in people living with HIV in the context of depression or medication adherence, few studies have included loneliness in such investigations; this could be a fruitful avenue for future research.

## Conclusion

The problem of loneliness in older men living with HIV is not principally an individual problem. Rather, experiences of loneliness for older gay men have a major societal element rooted in intersectional stigma. Yet, these men continue to bear the negative consequences. This study adds yet more evidence of the widespread impact of negative attitudes and beliefs about HIV on those living with this infection and underscores the need for societal intervention. This study also shows how older gay HIV+ men live in the face of these challenges, which can inform practical interventions at both individual and societal levels that could mitigate the quality-of-life-limiting experience of loneliness inclusive of others.

## Acknowledgments

The authors also extend their sincere appreciation to the participants who bravely shared their experiences.

## Author Contributions

**Conceptualization:** Marie-Josée Brouillette.

**Formal analysis:** Amanda Austin-Keiller, Melissa Park.

**Investigation:** Amanda Austin-Keiller.

**Methodology:** Melissa Park.

**Project administration:** Amanda Austin-Keiller.

**Supervision:** Melissa Park, Nancy E. Mayo, Marie-Josée Brouillette.

**Writing – original draft:** Amanda Austin-Keiller.

**Writing – review & editing:** Melissa Park, Seiyan Yang, Nancy E. Mayo, Lesley K. Fellows, Marie-Josée Brouillette.

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
