## [Decision Letter · Decision Letter 0]

14 Jun 2022

PONE-D-21-23275“Alone, there is nobody”: A qualitative study of the lived experience of loneliness in older men living with HIVPLOS ONE

Dear Dr. Brouillette,

Thank you for submitting your manuscript to PLOS ONE. After careful consideration, we feel that it has merit but does not fully meet PLOS ONE’s publication criteria as it currently stands. Therefore, we invite you to submit a revised version of the manuscript that addresses the points raised during the review process.

We look forward to receiving your revised manuscript.

Kind regards,

Limin Mao, PhD

Section Editor

PLOS ONE

Journal Requirements:

2. Please include a copy of the interview guide used in the study, in both the original language and English, as Supporting Information, or include a citation if it has been published previously.

Additional Editor Comments (if provided):

Dear Marie-Josée

Sorry for the delay in getting your paper for proper peer review. Apart from two reviewers' reports (both from Australia, I'm aware), I have read your manuscript myself. I appreciate its value in addressing some obvious gaps in HIV-related behavioral research.

Reviewers' comments:

Reviewer's Responses to Questions

**Comments to the Author**

1. Is the manuscript technically sound, and do the data support the conclusions?

Reviewer #1: Partly

Reviewer #2: Yes

2. Has the statistical analysis been performed appropriately and rigorously? 

Reviewer #1: N/A

Reviewer #2: N/A

3. Have the authors made all data underlying the findings in their manuscript fully available?

Reviewer #1: Yes

Reviewer #2: Yes

4. Is the manuscript presented in an intelligible fashion and written in standard English?

Reviewer #1: Yes

Reviewer #2: Yes

5. Review Comments to the Author

Reviewer #1: Thank you for the opportunity to review this manuscript. As an ageing cohort of people living with HIV enter into old age, the psycho-social aspects as much as the biomedical will need increasing attention. This manuscript is very welcomed in its focus on loneliness in older men living with HIV.

While the findings are generally robust and the analysis is sound, especially in the linking correlation of stigma across the themes, there are a few questions remaining:

1. The authors commence with a broad definition of loneliness as ‘as a negative feeling in response to a lower quantity or quality of social support than desired’, though there is little engagement with the broad sociology of loneliness. Instead, the authors rely on public health emphasises on health outcomes. Indeed, as the authors note at various points, an inductive thematic analysis proceeds from what the data says about loneliness, not how data fits into pre-existing frameworks. However, in order to make the claim that the psycho-social aspects of loneliness, especially in relation to stigma, the authors must engage with this significant body of sociological work. Else, more care need to be taken to ensure that the definition employed is not assumed in participants’ descriptions. Otherwise, I question which definition of loneliness was employed by research participants and how was loneliness narratives identified by researchers.

2. Relatedly, given the outsized role of stigma in both the findings and Discussion, the authors need to engage more with the literature to avoid taking stigma as a priori. For example, I read both interpersonal and anticipated/internalised stigma in some of the participants’ accounts. In the limitations, the authors were at pains to point to correlation not causation, yet, causation is implied throughout. A more careful engagement with the theory(ies) of stigma is needed to ensure that the layering of stigma over experiences and coping with loneliness to robust.

3. I applaud the authors for their innovation in their approach to member checking (usually confined to sending participants copies of transcripts or emergent findings). However, the dyadic nature of the second phase needs more careful attention. Akin to focus groups, this phase needs attention to both the thematic AND interactionalist content of those sessions. These would have a synergistic element, and I wonder how this interaction led to the refinement of the first theme, but also the suggestions for future interventions. Also, it was unclear whether some/all the quotes were derived from the first or second phase (e.g. page 13 quote). This needs to be clear upfront. How was data presented in pages 2? Anonymous? No quotes?

4. Associated with this, I had some questions around method. The authors mention what I relate as the coproduction of knowledge in the first phase, whereby narratives are presented and then subjectively interpreted by the interviewer. This is evident in the emphasis on significant, rather than cumulative, losses. How did the authorship team manage this, noting of course that this was listed as a limitation? More detail, then, is needed on an explanation of the benefits/challenges of narrative-phenomenology, as this is the first time I have heard of it together (conventionally, these methodologies are considered separate). What is this approach, other than a narrative-derived approach to analysing significant events? And if simply this, what makes this approach different to phenomenology?

5. The relationship of this sub study to the larger +BHN needs further explanation. Why were only 6 lonely and 4 non-lonely selected? Given the distinction between loneliness or no-loneliness, which quotes are attributed to which self-reported loneliness scale? How did +BHN findings also influence this analysis? How many were in original sample? How were they recruited? If by direct invitation, how was perceived coercion managed? There also needs to be further explanation of the scales selected for participant inclusion (eg why work, age, living with another, why not volunteering experience, intimate relationships, etc included?). I also found the table presentation very difficult, a simple X denoting yes or no would have been easier to read. What are the cumulative influences of these scales? Finally, I also wondered about the stage 1 questions. They are, as noted, very broad, but also an emphasis on ‘you’, rather than relational questions. I would be more comfortable with an explanation of how these questions were developed, and what role of the interviewer in prompts, follow-ups etc.

Minor notes:

- What is the role of the first author? All other authors have their titles (professor, and so on). If this is a student project, then this needs to be clear.

- Page 3, word missing after mental and physical?

- Page 3 citation needed for UK initiative or at least description of the strategy.

- Grammatical/spelling/missing words throughout that need proof reading.

- Page 4- ‘Research on HIV and aging, as well as on HIV and loneliness, has been dominated by quantitative studies (13), leaving a knowledge gap with respect to the lived experience of people with HIV. Qualitative research can fill this knowledge gap by focusing on subjects’ experiences and their own explanations of multivariable, complex difficulties”. Further explanation needed – what are the partialities of quant research? In which domains? What gaps precisely? What are the implications of developing interventions on the basis on quant studies alone?

- Page 8, in stage 2 description, what theory? Should this be findings?

- Page 10 – themes were populated. Surely derived? As populated denotes fitting in with existing theoretical framework

- Page 12 – information related to member checking and theme refinement better moved to that section.

- Page 13 – who is ‘me’? interviewer?

- Page 13 – no evidence for this claim ‘This constriction is even more pronounced as they and their friends age and, naturally, face more health difficulties”

- Page 32- ‘Many studies have investigated alcohol or drug-use in people living with HIV in the context of depression or medication adherence” refs needed

- Page 33 – ‘‘skeletal beings on their deathbed’. To whom is this quote attributed?

Reviewer #2: This paper reports on a study that explored the social phenomena of loneliness, an issue that is gaining increasing attention in the HIV sector in relation to health outcomes and quality of life for people living with HIV. This paper will be of interest to researchers and people working in community, support and clinical services. The paper is sound overall and worthy of publication. The method is really well considered and well implemented and the findings are well presented. The background and methods and discussion need to just be a bit tighter. I have put some suggestions below:

Introduction

Page 1, first paragraph. The definition of loneliness is very brief and refers only to social support. The references cited to support this definition offer a much more comprehensive definition of loneliness related to the absence, or limitations, of social networks, the quality of social relationships, trust and solidarity. It would be good if some of these themes are brought into the definition here as it is possible for people to be lonely even if they don’t require emotional or practical support as strictly defined.

Also, in paragraph two, loneliness is described as a feeling. Without wanting to be too finicky, I wonder if that is accurate? While I appreciate that loneliness is a feeling, and that emotions reflect and reinforce social processes, the policy and programmatic response to loneliness referred to above (eg. the UK government response) is really a response to loneliness as a social phenomenon. Again, a more detailed definition of loneliness might tease this out.

In this same paragraph, it would be helpful to provide a brief outline of the reasons why loneliness is associated with stigma. Have the studies referred to provided insight into this?

Methods

The section under the subheading ‘Qualitative approach’ is a little hard to follow. In general, it covers too much ground for a short paragraph and some concepts raised need more detail. For example, the statement ‘As specified by Charmaz (2006) in her approach to grounded theory, our findings are based on the experiences of our research participants as they are interpreted by the researcher’. More details on the subjective position of the researcher as this informed their approach to data analysis is needed to build from this statement. Also, more details on narrative-phenomenology and its person-centered, event-centered focus on experience is needed. Explain what this is an how it has been applied to this research.

Discussion

There are a few places where the meaning is not clear to the reader. For example, “The interviewer remembered being told about the multiple losses, but the stories of individual loss resonated with her more as they were closer to her own experience.” I think this means that it was difficult for the interviewer to understand the impact of multiple losses based on personal experience, but it reads as thought the researcher is reflecting on some specific elements of her life which would need explaining for the reader to understand how this shaped the analysis.

Also, the point that most people cannot fathom multiple losses and thus provide empathy needs some deeper consideration with respect to existing literature. I imagine this topic has been addressed in studies on loneliness and ageing given cumulative loss of friends and partners is very much an experience of ageing. I also know that there was work on grief and loss among gay men in relation to multiple losses from AIDS conducted in the 1990s that may be worth looking at here.

The discussion could be a bit shorter and more focused on implications rather than describing each element of the findings. A general edit should do this, however.

6. PLOS authors have the option to publish the peer review history of their article (what does this mean?). If published, this will include your full peer review and any attached files.

Reviewer #1: No

Reviewer #2: No

---

## [Author Response · Author response to Decision Letter 0]

24 Oct 2022

Thank you to the Reviewers for their extensive comments, which inspired a return to a more nuanced look at the data, refinement of headers, search for and integration of new references on intersectionality and stigma integrated into the discussion. Sincerely, Melissa Park.

As mentioned in the general comments, these changes made it difficult for us to keep working from a tracked changes version -- and one author accepted the changes to keep working. Therefor, we submitted a pdf of five versions of the rewrites. Please let us know if you would like to see specific sections/edits, and we can upload those as word docs with the tracked changes.

---

## [Editor Report · Decision Letter 1]

27 Oct 2022

“Alone, there is nobody”: A qualitative study of the lived experience of loneliness in older men living with HIV

PONE-D-21-23275R1

Dear Dr. Park,

We’re pleased to inform you that your manuscript has been judged scientifically suitable for publication and will be formally accepted for publication once it meets all outstanding technical requirements.

Kind regards,

Limin Mao, PhD

Section Editor

PLOS ONE

Additional Editor Comments (optional):

Thanks, Melissa and your co-author team, for taking such a good care in the revision. Great work on an extremely important topic!
---

## [Editor Report · Acceptance letter]

8 Dec 2022

PONE-D-21-23275R1 

“Alone, there is nobody”: A qualitative study of the lived experience of loneliness in older men living with HIV 

Dear Dr. Park:

I'm pleased to inform you that your manuscript has been deemed suitable for publication in PLOS ONE. Congratulations! Your manuscript is now with our production department. 

Kind regards, 

on behalf of

Dr. Limin Mao 

Section Editor

PLOS ONE